# Feasibility and Impact of Embedding an Extended DNA and RNA Tissue-Based Sequencing Panel for the Routine Care of Patients with Advanced Melanoma in Spain

**DOI:** 10.3390/ijms25136942

**Published:** 2024-06-25

**Authors:** Natalia Castrejon, Roberto Martin, Antonio Carrasco, Paola Castillo, Adriana Garcia, Raquel Albero-González, Mireia García, Marta Marginet, Núria Palau, Mónica Hernández, Carla Montironi, Guillem Clot, Ana Arance, Llucia Alos, Cristina Teixido

**Affiliations:** 1Department of Pathology, Hospital Clinic, University of Barcelona, 08036 Barcelona, Spainancarrasco@clinic.cat (A.C.); pcastillo@clinic.cat (P.C.); apgarcia@clinic.cat (A.G.); raalbero@clinic.cat (R.A.-G.); garcia01@clinic.cat (M.G.); mmarginet@clinic.cat (M.M.); mhernandez@clinic.cat (M.H.); montironi@clinic.cat (C.M.); lalos@clinic.cat (L.A.); 2August Pi i Sunyer Biomedical Research Institute (IDIBAPS), Rosselló 149, 08036 Barcelona, Spain; gclot@ub.edu (G.C.); amarance@clinic.cat (A.A.); 3Department of Medical Oncology, Hospital Clinic, University of Barcelona, 08036 Barcelona, Spain; rmartinh@clinic.cat; 4Molecular Biology Core Facility, Hospital Clínic, 08036 Barcelona, Spain; npalau@clinic.cat; 5Department of Basic Clinical Practice, Faculty of Medicine and Health Sciences, University of Barcelona, 08036 Barcelona, Spain

**Keywords:** melanoma, molecular pathways, *BRAF*, *NRAS*, next-generation sequencing, oncomine

## Abstract

Targeted NGS allows a fast and efficient multi-gene analysis and the detection of key gene aberrations in melanoma. In this study, we aim to describe the genetic alterations in a series of 87 melanoma cases using the oncomine focus assay (OFA), relate these results with the clinicopathological features of the patients, and compare them with our previous study results in which we used a smaller panel, the oncomine solid tumor (OST) DNA kit. Patients diagnosed with advanced melanoma at our center from 2020 to 2022 were included and DNA and RNA were extracted for sequencing. Common mutated genes were *BRAF* (29%), *NRAS* (28%), *ALK*, *KIT*, and *MAP2K1* (5% each). Co-occurring mutations were detected in 29% of the samples, including *BRAF* with *KIT*, *CTNNB1*, *EGFR*, *ALK*, *HRAS*, or *MAP2K1.* Amplifications and rearrangements were detected in 5% of cases. Only *BRAF* mutation showed a significant statistical association with sun exposure. For patients with a given genetic profile, the melanoma survival and recurrence-free survival rates were equivalent, but not for stage and LDH values. This expanded knowledge of molecular alterations has helped to more comprehensively characterize our patients and has provided relevant information for deciding the best treatment strategy.

## 1. Introduction

Melanoma accounts for 1% of all skin cancers, but it is the most serious and deadly type of skin cancer [1,2]. Melanoma incidence has dramatically increased in the last 20 years, mainly due to increasing levels of ultraviolet exposure, especially in people over the age of 60, and will continue to grow for decades. [3] In Spain, the annual incidence rate of cutaneous melanoma per 100,000 people has increased in both men and women, by 2.5% and 1.6%, respectively [4].

The prognosis of advanced staged melanoma (AM), involving stage III and IV, has significantly improved recently due to a better knowledge of melanoma molecular pathways and the development of immunotherapy and targeted treatments [5,6]. However, AM patients’ management is still a challenge due to melanoma genetic complexity, and its capacity to elude the immune system and to develop drug resistances [7,8].

Approximately 40 to 50% of AM harbor *BRAF* V600 mutation, and the affected patients can benefit from BRAF and MEK inhibitor treatment when the mutation is identified [9,10,11]. Therefore, *BRAF* mutational status determination is mandatory in AM patients due to its predictive value of response to BRAF and MEK inhibitors, although these patients can also benefit from immune checkpoint inhibitors (ICIs) [12].

On the other hand, in wild-type melanoma, anti-BRAF+/−MEK therapies are not indicated, and ICIs are the treatments that have been demonstrated to improve overall survival. Nevertheless, the treatment of this subset of patients is still challenging. Moreover, wild-type *BRAF* in AM harbor diverse mutations, of which rates and implications have not been completely elucidated [13,14].

Although the majority of these patients experience clinical benefit from BRAF and MEK inhibitors, most patients with *BRAF* V600-mutated metastatic melanoma develop resistance to these agents [15]. Different genetic mechanisms of *BRAF* resistance have been identified including secondary mutations of *BRAF* V600, MAPK pathway reactivation, or alternative survival pathways’ activation [16,17,18]. There are still limitations in their understanding which is fundamental to decide the correct line of research.

Although the real-time polymerase chain reaction (RT-PCR) technique is the most widely used analysis to study the *BRAF* gene status, it seems reasonable to use multi-gene panel analysis strategies, such as targeted next-generation sequencing (NGS), when available, to obtain additional genetic data, which can be useful for melanoma characterization.

Our hospital is one of the referral centers in Spain for the management and treatment of patients affected by melanoma and has a melanoma unit composed of an experienced multidisciplinary team. In this setting, our pathology department is equipped with the most innovative technologies to perform a diagnosis of excellence including NGS techniques. In 2020, we replaced the oncomine solid tumor (OST) 22-gene hotspot DNA panel with the oncomine focus assay (OFA) DNA + RNA panel, a larger panel that allows the study of 52 gene alterations, including single nucleotide variants, insertions, deletions, amplifications, and rearrangements of key genes. In view of the fact that the OFA includes some relevant genes in melanoma pathways apart from *BRAF*, such as *NRAS*, *KIT*, or *MAP2K1*, we embraced this panel to prospectively study a series of advanced melanoma patients.

The main aim of this study was to provide detailed genetic information of a Spanish series of AM patients using a 52-gene panel assay, assess the associations between the molecular findings and clinical parameters, and compare these results with our previous data [19].

## 2. Results

### 2.1. Patient Characteristics

A total of 87 patients were included in the study. Patients’ median age was 61 years (a range from 8 to 91 years), including only four patients under 40 years old. Forty-five out of eighty-seven (51.7%) patients were female and forty-two (48.3%) were male (see Table 1). In fifty-two out of seventy (74%) patients with available information, the melanoma site was located in a sun-exposed region, whereas in eighteen patients the neoplasm was located in a sun-shielded area. Four patients presented with metastatic disease with no evidence of primary lesion.

Out of a total of 80 patients, the majority were at an advanced stage when tested (76/80; 95%), with 23 (28.8%) at stage III and 53 (66.2%) at stage IV. Among the latter, seven had central nervous system (CNS) involvement (stage IV M1 D). Serum lactate dehydrogenase (LDH) levels were elevated in 16 patients.

### 2.2. Histological Features

All samples selected for the molecular analyses were evaluated with at least a hematoxylin and eosin stain in which the percentage of neoplastic cells, as well as the tumor area, were assessed. The percentage of neoplastic cells in the samples ranged from 20% to 95%, with an average of 77% in the whole series.

Regarding data of primary melanoma, most of the patients had superficial spreading melanoma subtype (n = 24, 28%), followed by nodular melanoma (n = 15, 17%), acral lentiginous melanoma and mucosal melanoma (n = 8 each, 9%), lentigo maligna melanoma (n = 6, 7%), nevoid melanoma (n = 3, 3%), and spitzoid and uveal melanoma (n = 2, 2%, each) (Table 1). Subtype classification was not possible to establish in four cases because the primary melanoma was not clinically found. In 15 cases the pathological report of primary tumour evaluation did not specify the melanoma subtype and these data were not possible to assess in the histological evaluation.

The average of the Breslow index was 2.2 mm, ranging from 0.5 to 18 mm. Ulceration was found in 50% of the cases with available information (29/58, 50%).

### 2.3. Somatic Mutations Detected by NGS

A total of 87 FFPE melanoma samples were submitted for molecular studies. All samples were derived from either biopsy or excision of primary melanoma (n = 34) and from biopsies or resections of metastasis (n = 53).

*BRAF* mutation was the most frequent alteration found in our series, identified in 24 out of 82 (29%) patients with assessable results. *NRAS* mutation was found in 23 patients (28%). Other mutated genes were *ALK*, *KIT*, and *MAP2K1*, which were identified in four patients (5% each) (Figure 1).

Regarding *BRAF*-mutated cases, superficial spreading melanoma (SSM) was the most frequent subtype (11/24, 46%). Among these cases, the Breslow index average was 3.1 mm and 44% of the cases with available information were ulcerated.

Considering *NRAS*-mutated cases, the most prevalent subtypes were SSM (5/23, 22%), and lentigo maligna melanoma (3/23, 13%). The Breslow index average was 4.9 mm and 10 out of 17 (59%) were ulcerated.

*BRAF*, *NRAS*, *KRAS*, *ERBB3*, *HER2*, *JAK1*, *PIK3CA*, and *SMO* mutations, and *MYC*, *CCND1*, and *KIT* amplifications were mutually exclusive. On the other hand, co-occurrence of oncogenic mutations was detected in 22 assessable cases (25.3%). In detail, 10 of the *BRAF*-mutated cases had a concurrent alteration in *ALK* (two cases), *CTNNB1* (two cases), *EGFR* (two cases), *HRAS* (one case), *KIT* (one case), and *MAP2K1* (two cases, one of them together with *AKT1*).

Considering *NRAS*-mutated cases in combination with other alterations, we found eight cases which had alterations in other genes. Among them, one had an *ALK* mutation together with a *PDGFRA* mutation, and one *ERBB4* with *FGFR4* amplification, and five cases harbored mutations in *KIT*, *IDH2*, *IDH1*, and *JAK2* (2 cases). One of the patients with *NRAS* and *JAK2* mutations had myeloid leukemia with a prior described *JAK2* mutation. Moreover, one case showed a rearrangement of *ETV1* with *PTPRZ1*. In total, four cases had three concurrent alterations and one case had four aberrations (see Figure 1).

The two statistical associations found were between the presence of *BRAF* mutation and sun exposure, and *ALK* alterations with gender (see Table 2).

### 2.4. Amplifications and Rearrangements Detected by NGS

Scarce gene amplifications or rearrangements were identified in our series, n = 5 and n = 2, respectively. In detail, *MYC* amplifications were found in two (2%) cases, and *CCND1*, *FGFR4*, and *KIT* in one case each (1% each), whereas *ALK* and *ETV1* rearrangements were found in one case each (2%, each) (Figure 1). The low rate of these types of alterations limited the statistical analysis, so it was not possible to find associations with clinical or histological variables.

### 2.5. Concordance between RT-PCR and NGS Techniques for BRAF Mutation

The clinical management implications of NGS and RT-PCR results diverge in our routine clinical practice. RT-PCR plays a crucial role in guiding our therapeutic approach, while NGS results influence decisions related to inclusion/exclusion in specific clinical trials. For instance, NGS aids in identifying patients with RT-PCR wild-type tumors who lack alternative therapeutic options or helps elucidate the specific *BRAF* V600 mutation type. This distinction is essential for tailoring effective and targeted treatments based on individual patient profiles. The OFA was validated using the *BRAF* RT-PCR test, as it is the standard routine diagnostic tool in our institution.

A total of 55 samples were tested by both methods: 17 from biopsy or excision of primary melanoma and 38 from biopsies or resections of metastasis. A high concordance rate of 88% was observed among 55 out of 87 samples that underwent evaluation through both methodologies, yielding a Cohen’s kappa value of 0.72.

Among the samples tested, sixteen were mutated in *BRAF* as identified by RT-PCR. These mutations comprised of thirteen samples of V600E/E2/D and three of V600K/R/M. In parallel, NGS revealed *BRAF* alterations in eighteen samples. The most frequent *BRAF* mutation identified by NGS was V600E (19; 79%), followed by V600K (2; 9%). Other less frequent mutations included G596R, G466, and V600_K601delinsD, each observed in a single case (4% each).

Six out of fifty-five cases were discordant, five of them were not identified by using the RT-PCR method, whereas one of them was not detected by the NGS technique. Three out of the five *BRAF*-mutated cases non-detected by RT-PCR harbored mutations in codon 600: in detail, V600E and V600K mutations were identified in two cases and one case, respectively. The remaining two cases showed alterations in *BRAF* gene codons different than 600, specifically, G596R and G466E mutations. On the other hand, in the case in which the OFA did not show *BRAF* mutation, the RT-PCR result indicated V600E/V600E2/V600D. Discordant cases were tested twice. Since RT-PCR is not able to capture non-*BRAF* V600 mutations, the real correlation between the two techniques was in fact 92.7% (51/55).

### 2.6. Treatment Strategies and Overall Survival

Out of the total cohort, 72 patients were treated, 7 did not receive any treatment, and treatment details were unavailable for 8 patients. Immunotherapy was administered to a total of 66 patients, with 23 receiving it as part of an adjuvant regimen and 43 in the first-line treatment. The adjuvant regimen primarily consisted of anti-PD1 monotherapy. One patient in this group received a combination of anti-PD1 with an anti-CTLA4. In the context of first-line treatments, 2 patients underwent anti-CTLA4 monotherapy, 28 received anti-PD1 monotherapy, and 13 were treated with an anti-PD1-based combination (11 with anti-CTLA4 and 2 with anti-LAG3). Targeted therapy was given to four patients (one in adjuvant and three in first-line treatment), and two patients were administered first-line chemotherapy (dacarbazine). A significant PFS, but not OS, was observed in patients treated with immunotherapy compared to those receiving other types of treatments (*p* = 0.030), Table 3.

In terms of follow-up, 52 patients are currently alive. Among them, 20 are in complete response, 11 in partial response, and information about response status is not available for 6 patients. However, 28 patients relapsed and died.

Kaplan–Meier analysis with log-rank testing showed melanoma-specific survival was lower for stage IV D (*p* = 0.049) and elevated LDH levels (*p* < 0.001) (Figure 2). Differences were observed between stage IV subtypes and stage III (Table 3). Of those who died, mortality occurred sooner in the stage IV D cohort, with a median of 33.76 months until mortality versus 133.47 months until mortality in the stage III cohort.

## 3. Discussion

The implementation of techniques to identify specific genetic alterations that drive melanoma development and progression is crucial for selecting specific treatments, as well as for the development of potential targeted therapy and the prediction of patient response to treatment.

NGS is a high-throughput technology able to detect genetic alterations simultaneously, including mutations, copy number variations, and gene fusions, with high sensitivity and specificity. NGS gene commercial panels include a variety of cancer-related genes designed to cover different needs. The OFA has been specifically designed to provide comprehensive genomic profiling of solid tumors and includes the mutational study of melanoma-related genes such as *BRAF*, *NRAS*, *KIT*, *MAP2K1*, or *HRAS*. In the previous study performed by our group [19], we proved the feasibility of implementing an NGS assay using a 22-gene panel to study melanoma samples of patients in advanced stages. In this context, we took advantage of the acquisition of the OFA in our department and performed a retrospective study of a series of 87 melanoma cases from 2020 to 2022.

In our series, cutaneous melanoma was the most common type, although mucosal and uveal melanomas were also included. The prevalence of *BRAF* mutations in the total cohort detected by NGS was 29%, which is considerably low compared to those reported in the literature, including the study performed by our group [19,20,21,22]. This can be explained by two main reasons. Lately, we have been seeing older patients, and many of these patients tend to be wild type (WT) [23]. On the other hand, in our institution, the OFA is usually used in *BRAF* WT patients by Idylla (RT-PCR) with the aim of identifying molecular alterations that may serve for potential therapies, either within or outside of a clinical trial. In other words, there is an enrichment in this WT series, as these are the cases for which we would request NGS.

Moreover, the number of patients studied by an NGS panel in our department during the two-year period of this study (2020–2022) diminished significantly compared to the period of 2017 and 2019 that we published [19], being 87 and 155, respectively. This is because previously we performed both tests on patients who had sufficient material to gain experience in conducting and assessing results through NGS. As mentioned above, nowadays, we mostly conduct NGS on BRAF WT patients using RT-PCR, which allows us to reduce costs and save personnel time for the implementation of other techniques.

Twenty-eight percent of our cases harbored a *NRAS* mutation, which is slightly higher than that described in previous studies [21,24,25]. A remarkable finding was that *NRAS* mutations’ percentage increased from 14% in 2017–2019 to 28% in the 2020–2022 period. This can be explained as commented above because our population study was enriched with *BRAF* WT. On the other hand, TP53 was not included in the Oncomine Focus gene panel, which constituted the second most frequent mutation in our previous study. In contrast, the OFA, but not OST, includes *KIT* gene, a key player in the pathogenesis of melanoma, especially in acral and mucosal melanoma with a frequency of 1–7% [26,27,28,29]. Among our four cases with *KIT* mutation (5% of prevalence), one was from a mucosal site, whereas the others were cutaneous tumors, one of which was classified as an acral melanoma. None (0%) were identified in melanomas on skin without chronic sun damage. Remarkably, three cases had concomitant mutations, including *BRAF* and *NRAS* genes. Notably, we identified four cases (5%) with *ALK* mutation, all of which were located in sun-exposed areas and had a Breslow index superior to 1.5 mm, and one of the patients had xeroderma pigmentosum. Regarding the four *MAP2K1* mutated cases, three of them had concurrent mutations, one with *BRAF*, one with *BRAF* and *AKT1*, and one with *ROS1*.

The implementation of the OFA allowed not only the identification of hotspot alterations, but also reported rearrangements and amplifications. In this study, the OFA helped to identify two rearrangements and seven amplifications. Although *BRAF* V600 mutations are usually mutually exclusive with other driver alterations, in our series, a case of metastatic nodular melanoma presented a *BRAF* V600E co-mutation with an *ALK* rearrangement. Both were confirmed by another orthogonal molecular technique, RT-PCR and fluorescence in situ hybridization, respectively. However, the clinical management of this patient has not changed at present. As yet, studying advanced melanoma samples through a larger but not specifically selected gene panel has not changed the clinical management of these patients.

A concordance of 92.7% was observed between RT-PCR and NGS studies regarding *BRAF* V600 status, in agreement with previous studies [30]. There were four discordant cases: three of them were not detected by the RT-PCR technique and one was not identified by NGS. Despite the turnaround time and cost of the RT-PCR being lower compared to the NGS method, the second technique is the preferred in selected cases which entail complex clinical management or difficult histological diagnosis, since it provides the specific *BRAF* V600 alterations present, and it delivers additional information beyond mutations in the *BRAF* V600 locus, which can be decisive in these contexts. Moreover, guidelines for negative cases of *BRAF* class I alterations recommend sequencing other less frequent mutated loci (BRAF class II and III) to verify *BRAF* wild-type status [31], as well as to analyze *NRAS* and *KIT* genes, all covered with the OFA [32,33]. One explanation of these discordance results in three cases is that different samples were used for one technique and another. Moreover, the high concordance observed, and the detailed characterization of mutations, provide valuable insights into the molecular profile of the tested samples, contributing to the precision and reliability of diagnostic assessments in our institution.

Importantly, using the OFA has increased the detection of a large number of specific molecular alterations in AM, in comparison with the results of our previous study using the OS panel. Nevertheless, the OFA still lacks some important genes involved in melanoma development such as *NF1*, *CDKN2A*, *TP53*, or *TERT* [21,34], which is a noticeable limitation of our study. The lack of a commercial NGS assay for melanoma assessment drove us to the next step of designing a customized gene panel for melanoma, which is currently an ongoing project of our research group. This expanded knowledge of molecular alterations in advanced melanoma could be the basis for the development of novel targeted treatments.

## 4. Materials and Methods

### 4.1. Patient and Samples

Between June 2020 and March 2022, all patients newly diagnosed with advanced melanoma in our institution were included in the study. Prior full informed consent of the patients and approval from the Internal Review Board of the Hospital Clinic of Barcelona (Barcelona, Spain; HCB/2017/0097) were obtained. The study was conducted following the principles of the Declaration of Helsinki.

Before any test was performed, formalin-fixed and paraffin-embedded (FFPE) tissue blocks and hematoxylin and eosin-stained whole slides from all patients were collected and evaluated in order to estimate the percentage of neoplastic cells, as well as tumor area. Samples with tumor cell content higher than 20% and with a tumor area of at least 4 mm^2^ were considered assessable for sequencing analysis. This initial sample evaluation was carried out by N.C., A.C., L.A., and C.T.

Clinical data and pathological reports were retrieved from the electronic medical records. Considering that our institution is a referral center for adult patients affected by melanoma, our series was mainly composed of melanoma samples received from other hospitals and a predominantly adult population.

### 4.2. Targeted NGS

For each FFPE tumor sample, cut sections were collected for nucleic acid extraction. DNA (QIAamp DNA FFPE Tissue Kit; Qiagen, Hilden, Germany) and RNA (High Pure FFPET RNA Isolation kit; Roche Diagnostics, Mannheim, Germany) extraction were performed following the manufacturer’s instructions and quantified using a Qubit Quantitation Assay Kit in a Qubit Fluorometer (Life Technologies, Gaithersburg, MD, USA) [35].

An input of 10 ng of DNA and RNA were amplified using the NGS targeted panel OFA (ThermoFisher Scientific, Waltham, MA, USA) following the manufacturer’s instructions. The panel was designed to identify 35 genes with hotspot mutations, 19 genes with focal copy number variations (CNVs) gains and 23 genes with fusion drivers (see Table 4 and Figure 3). The sequencing was performed with an Ion S5 platform and analyzed with the Torrent Suite program v5.8, and variants detected were annotated and filtered with the Ion Reporter software v5.14.

### 4.3. RT-PCR for BRAF V600 Mutational Status Determination

Cut sections were obtained from FFPE samples to perform the in vitro automated Idylla^TM^
*BRAF* Mutation Test (Biocartis, Mechelen, Belgium), which allowed the qualitative detection of V600E/E2/D and V600K/R/M mutations in codon 600 of the *BRAF* gene. We obtained 5 to 10 μm tissue sections of samples with a tumor cell content between 10 to 90% and a neoplastic area superior to 2 mm^2^ following the manufacturer’s instructions.

### 4.4. Statistical Analysis

For each gene, the Fisher’s exact test or the Mann–Whitney U test was used to evaluate the association between the presence of alterations and categorical or continuous variables, respectively. Mutations, amplifications, and rearrangements were considered as alterations. The Mann–Whitney U test and the Spearman correlation were used to evaluate the association between the total number of alterations and categorical or continuous variables, respectively. *p*-values were adjusted using the Benjamini–Hochberg method. Progression-free survival and overall survival from the time of treatment were used as endpoints. Survival curves were estimated with the Kaplan–Meier method and compared with the log-rank test, except for the survival curves according to the Stage variable. Those were compared with the test for trend implemented in the survMisc R package. Hazard ratios were estimated with Cox regression. All statistical tests were carried out with R version 4.3.0.

## Figures and Tables

**Figure 1 ijms-25-06942-f001:**
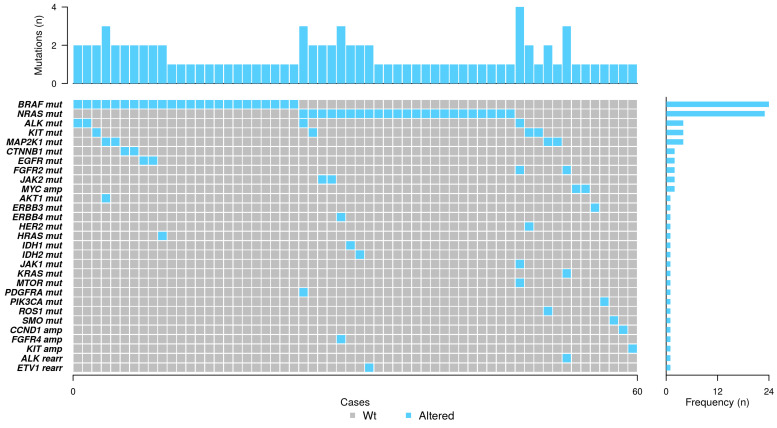
Co-mutation plot of genetic alterations identified by next-generation sequencing. Spectrum of genes (rows) with alterations identified in 87 advanced melanoma cases (columns). Sixty cases are mutated. The percentage of samples with an alteration detected is depicted in the right histogram. Altered genes are highlighted in blue, while grey indicates those analyzed cases with no identified alterations. mut, mutation; amp, amplification; rearr, rearrangement; Wt, wild type.

**Figure 2 ijms-25-06942-f002:**
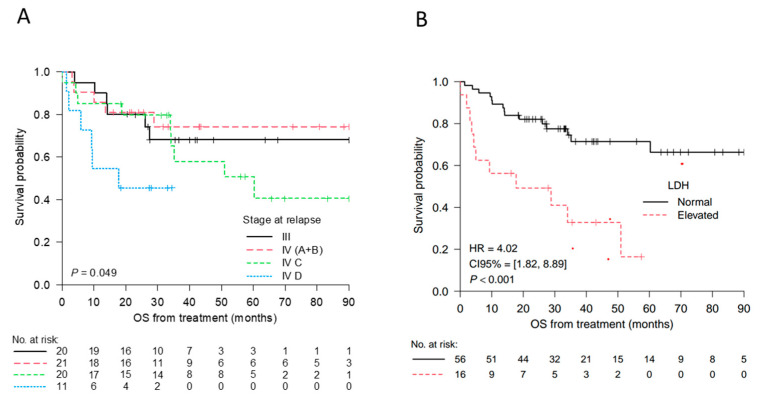
Kaplan–Meier curves for the overall survival (OS) from the time of treatment according to the stage at relapse (**A**), OS from the time of treatment according to the LDH levels (**B**). Numbers at risk are indicated at selected time points. Stage III: no distant metastasis (regional lymph node metastasis, in-transit or satellite cutaneous metastasis (M1). Stage IV A+B: A, Distant metastasis to skin, soft tissue including muscle and/or non-regional lymph node; B, lung metastasis. Stage IV C: distant metastasis to non-central nervous system (CNS) visceral sites with or without M1 A or M1 B sites of disease. Stage IV D: distant metastasis to CNS with or without M1 A, M1 B, or M1 C sites of disease. LDH elevated ≥234 U/L. The hazard ratio (HR) and the *p*-value (P) from the log-rank test comparing the two groups are shown.

**Figure 3 ijms-25-06942-f003:**
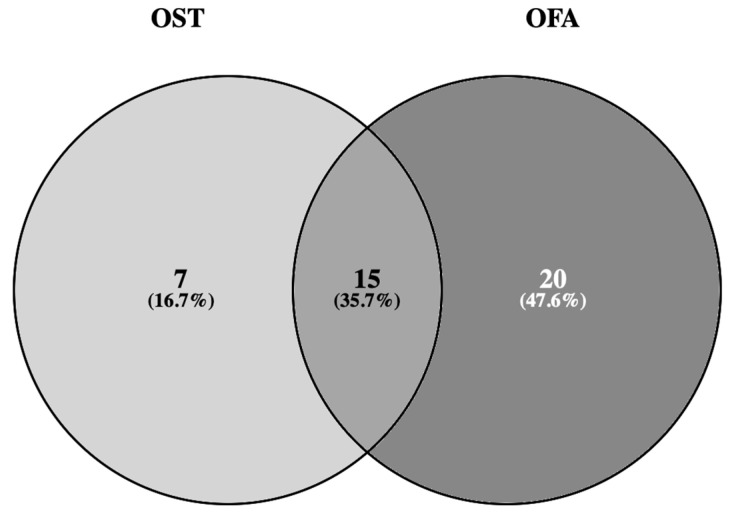
Venn diagram showing that 15 hotspot genes overlap between both the oncomine solid tumor (OST) and oncomine focus assay (OFA) [36], accessed on 23 December 2023.

**Table 1 ijms-25-06942-t001:** Clinicopathological features of 87 patients with melanoma investigated with the sequencing panel oncomine focus assay.

Parameter	Number of Cases
Mean age (range) (years old)	61 (8–91)
Sex	
Female	45/87 (51.7%)
Male	42/87 (48.3%)
Type of melanoma	
Superficial spreading melanoma	24/87 (28%)
Nodular melanoma	15/87 (17%)
Acral melanoma	8/87 (9%)
Mucosal melanoma	8/87 (9%)
Lentigo maligna melanoma	6/87 (7%)
Nevoid melanoma	3/87 (3%)
Spitzoid melanoma	2/87 (2%)
Uveal melanoma	2/87 (2%)
Not established *	19/87 (22%)
Melanoma localization	
Sun-exposed region	52/70 (74%)
Sun-shielded region	18/70 (26%)
Stage	
I	2/80 (2.5%)
II	2/80 (2.5%)
III	23/80 (28.8%)
IV	53/80 (66.2%)
Treatment	
Chemotherapy	2/79 (2.5%)
Immunotherapy	66/79 (83.5%)
Targeted therapy	4/79 (5.1%)
No treatment	7/79 (8.9%)

* Due to unknown primary melanoma, or non-received data from external center.

**Table 2 ijms-25-06942-t002:** Relation of clinicopathological characteristics with the most prevalent somatic alterations identified by next-generation sequencing.

Clinicopathological Characteristics	*BRAF* *(p)*	*NRAS (p)*
Sex		
Male	0.222	1
Female		
Age		
Median	0.186	0.514
NGS Site of biopsy		
Primary	0.134	0.611
Metastasis		
UV exposure		
Elastosis	0.013	0.758
Absence of elastosis		
Depth of invasion		
Median	0.286	0.454
Ulceration		
Absent	0.775	0.381
Present		
Vascular Invasion		
Absent	1	1
Present		

**Table 3 ijms-25-06942-t003:** Correlation between overall survival (OS) and progression-free survival (PFS).

	OS	PFS
LDH Normal (<234U/L) Elevated (≥234 U/L)	0.0001	0.124
Stage at relapse III IV	0.048	0.368
Type of treatment Anti-PD1 monotherapy Anti-PD1 combination (anti-CTLA4/anti-LAG3) Targeted therapy Other treatments	0.312	0.030
Treatment intention Adjuvant Metastatic	0.230	0.444
* BRAF* mutation Absent Present	0.556	0.343
* NRAS* mutation Absent Present	0.771	0.861
* BRAF* treatment interaction	0.320	0.031

**Table 4 ijms-25-06942-t004:** List of genes included in both oncomine focus and oncomine solid tumor panels.

Oncomine Gene Panels			
Hotspot Genes		Copy Number Variants	Fusion Drivers
*AKT1*	*JAK3*	*ALK*	*ABL1*
*ALK*	*KIT*	*AR*	*ALK*
*AR*	*KRAS*	*BRAF*	*AKT3*
*BRAF*	*MAP2K1*	*CCND1*	*AXL*
*CDK4*	*MAP2K2*	*CDK4*	*BRAF*
*CTNNB1*	*MET*	*CDK6*	*EGFR*
*DDR2*	*MTOR*	*EGFR*	*ERBB2*
*EGFR*	*NRAS*	*ERBB2*	*ERG*
*ERBB2*	*NOTCH1*	*FGFR1*	*ETV1*
*ERBB3*	*PDGFRA*	*FGFR2*	*ETV4*
*ERBB4*	*PIK3CA*	*FGFR3*	*ETV5*
*ESR1*	*PTEN*	*FGFR4*	*FGFR1*
*FBXW7*	*RAF1*	*KIT*	*FGFR2*
*FGFR1*	*RET*	*KRAS*	*FGFR3*
*FGFR2*	*ROS1*	*MET*	*MET*
*FGFR3*	*SMAD4*	*MYC*	*NTRK1*
*GNA11*	*SMO*	*MYCN*	*NTRK2*
*GNAQ*	*STK11*	*PDGFRA*	*NTRK3*
*HRAS*	*TP53*	*PIK3CA*	*PDGFRA*
*IDH1*			*PPARG*
*IDH2*			*RAF1*
*JAK1*			*RET*
*JAK2*			*ROS1*

Green indicates those genes which were sequenced to identify mutations in both panels, while yellow indicates those genes only studied in the oncomine solid tumor panel. Non-colored genes were only present in the oncomine focus assay.

## Data Availability

The raw data supporting the conclusions of this article will be made available by the authors on request.

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
