# Peer review of "Feasibility and Impact of Embedding an Extended DNA and RNA Tissue-Based Sequencing Panel for the Routine Care of Patients with Advanced Melanoma in Spain"

_ijms, 2024, doi:10.3390/ijms25136942_

Round 1

Reviewer 1 Report

Comments and Suggestions for Authors

This study describes the genetic alterations in a series of eighty-seven melanoma cases using the oncomine focus assay (OFA), relates these results with the clinicopathological features of the patients and compares them with their previous study results in which they used a smaller panel, the oncomine solid tumor (OST) DNA kit. The study highlights two key findings: Targeted NGS using the OFA assay can improve identification of patients for targeted therapies; BRAF mutation is associated with sun exposure. These findings are significant for both diagnosis and treatment of melanoma.

The manuscript is written clearly and understandably, using appropriate language. The structure is well-organized and logical, making it easy to understand the contents. Although not all bibliographical references are the most recent, they are still relevant to the topic covered and demonstrate an in-depth knowledge of the scientific literature. The research presented in the manuscript appears valid and well-conducted. The methodology used is described in detail and clearly, allowing the reader to understand the procedures. The figures are clear and easily readable, making it easier to understand the data presented. The conclusions of the manuscript are consistent with the data presented. Suggestions: Regarding the design of a specific panel for melanoma, it would be advisable to integrate already tested panels to be able to make comparisons with other studies. Oncomine panels rely on specific probes or primers to amplify and detect target genes. These panels have limited capacity in terms of the number of genes that can be analyzed simultaneously. Adding genes not present in the panel may require the removal of other genes, with potential implications for the analysis.

Reviewer 2 Report

Comments and Suggestions for Authors

The text is well-written, with clear and concise language. Technical terms are used appropriately, reflecting the academic nature of the study. The structure of the document, including the sections and subsections, is logically organized . I would revise the layout of Tables 1, 2, and 4 to be able not to break the figure into two pages thus making it easier to understand. Figure 1 appears to be cut off at the left side and figures 2b and 2c appear to be blurred compared to 2a.

Although the study introduces new data, the methodology and context are not completely new. The use of NGS for genetic analysis of melanoma is already well established and used routinely in many hospital centers. The court of patients used for statistical analysis, as mentioned in the manuscript, is altered by BRAF WT enrichment affecting the result of statistical analysis. I would advise the authors to expand the patients court thus obtaining a more robust and translatable result also in accordance with the published literature.

Would advise the authors to revise Tables 2 and 3 as they are difficult to interpret in some points. For example, in Table 2 the association between gender and ALK is not clear whether the incidence of this alteration/mutation is found more in men and less in women or vice versa. The same problem is found in Table 3 for LDH, or the types of treatments associated with OS and PFS. In Figure 2 in the Kaplan-Meier analysis, the stage IV group is divided into courts A, B, C and D but neither in the 2.6 section nor in the figure caption is clear the differences between the groups. Only in section 2.1 the stage IV court D is linked to CNS involvement but for groups A B and C there is no description. I would advise the authors to add a legend to make Figure 2 clearer and more comprehensible. I would suggest to the authors to delete/revise Figure 2c, the relationship between BRAF mutational status and OS in a court of patients enriched in WT might be altered.

The study presents valuable insights into the genetic landscape of advanced melanoma. However, the article would benefit from addressing the weaknesses identified, particularly by expanding the size of the cohort, ensuring a more balanced representation of BRAF mutational status. As mentioned in the conclusion of the paper, expanding a more melanoma-specific panel would certainly lead to strengthening the originality of the work done and have a potential impact on the clinical management of these patients.

Round 2

Reviewer 2 Report

Comments and Suggestions for Authors

Table 3 is still unclear to me..the value given to which condition refers? (example: LDH normal or elevated?).

However, in the complex, the work can be accepted.

Author Response

Response to Reviewer 2

Comments 1: Table 3 is still unclear to me..the value given to which condition refers? (example: LDH normal or elevated?). However, in the complex, the work can be accepted.

Response 1: Thank you very much for your comment. I apologize for not providing the LDH cutoff in the previous revision. Here are the levels used in our center and how we categorize LDH.

Normal levels of LDH in our laboratory are defined as follows:

  • Normal levels: less than 234 U/L
  • Elevated levels: 234 U/L or higher

Figure 2 and Table 3 have been modified and now specify the LDH cutoff levels.